

# Advances in macrophage and T cell metabolic reprogramming and immunotherapy in the tumor microenvironment

Hua Cheng and Yongbin Zheng

Department of Gastrointestinal Surgery, Renmin Hospital of Wuhan University, Wuhan, Hubei, China

## ABSTRACT

Macrophages and T cells in the tumor microenvironment (TME) play an important role in tumorigenesis and progression. However, TME is also characterized by metabolic reprogramming, which may affect macrophage and metabolic activity of T cells and promote tumor escape. Immunotherapy is an approach to fight tumors by stimulating the immune system in the host, but requires support and modulation of cellular metabolism. In this process, the metabolic roles of macrophages and T cells become increasingly important, and their metabolic status and interactions play a critical role in the success of immunotherapy. Therefore, understanding the metabolic state of T cells and macrophages in the TME and the impact of metabolic reprogramming on tumor therapy will help optimize subsequent immunotherapy strategies.

## INTRODUCTION

Tumor immunotherapy has brought new treatments and hope to cancer patients and has been successfully used in a wide range of tumor types. While some patients respond well to immunotherapy and experience ideal therapeutic effects, the majority unfortunately do not. One significant factor contributing to this lack of response is the dysregulation of T cells in the tumor microenvironment (TME), which can be irreversible (*Pauken et al., 2016*; *Philip et al., 2017*). The lack of effector function and the upregulation of inhibitory receptors, such as programmed cell death-1 (PD-1), T cell immunoglobulin and mucin domain-3 protein (TIM-3), lymphocyte-activation gene 3 (LAG-3) and CD39, are signs of T cell failure in tumor-infiltrating T cells that are exhaustion (*McLane, Abdel-Hakeem & Wherry, 2019*; *Miller et al., 2019*; *Schietinger et al., 2016*). T cell dysfunction is caused by antigenic stimulation that lasts for a long time and is affected by components outside of T cells in the TME, such as cytokines, chemokines, and metabolites from the immunosuppressive population in the periphery (*Thommen & Schumacher, 2018*; *Xia et al., 2019*).

Corresponding author
Yongbin Zheng,
yongbinzheng@whu.edu.cn

Antigen-presenting myeloid cells within the TME have the potential to hinder T cell function. Myeloid cells, such as macrophages and monocytes, are highly prevalent in most types of tumors and are believed to be the dominant population responsible for inhibiting T cell function (*DeNardo & Ruffell, 2019*; *Lavin et al., 2017*). Macrophages and monocytes may overexpress programmed death ligand 1 (PD-L1) and other ligands that interact with inhibitory receptors on T cells, leading to apoptosis (*Cassetta et al., 2019*; *Tang et al., 2018*). In addition, they can inhibit T cell function by producing regulatory cytokines, such as interleukin 10 (IL-10) and transforming growth factor β (TGF-β) (*Noy & Pollard, 2014*; *Speiser, Ho & Verdeil, 2016*), tumor-associated macrophages (TAMs) demonstrate high plasticity, with transcriptional regulation and epitranscriptional reprogramming coordinating specific gene expression patterns and determining phenotypic shifts in TAMs, leading to functional heterogeneity with distinct phenotypes (*Locati, Curtale & Mantovani, 2020*). However, the underlying mechanisms by which macrophages control T cell fate are not fully understood. Recent research shows that macrophages lacking Mettl14 or Ythdf2 have increased levels of Epstein-Barr virus-induced protein 3 (Ebi3) transcripts. This interferes with the infiltration of cytotoxic T cells and impairs the function of CD8[+] T cells, hampering their ability to kill tumors (*Dong et al., 2021*). Studies using *in vivo* imaging have revealed that CD8[+] T cells, specific to antigens, tend to congregate in regions of the TME that are heavily populated by TAMs (*Boissonnas et al., 2013*; *Broz et al., 2014*). The interactions formed between these cells are tight and persistent, as demonstrated by epigenetic and transcriptional analyses. Interestingly, depleted T cells are found to produce factors that participate in recruiting monocytes to the TME, affecting their differentiation. Dot matrix light-sheet microscopy has uncovered a novel and extended interaction between TAMs and CD8[+] T cells, which is antigen-specific and synaptic in nature without activating T cells. This interaction can lead to the depletion of T cells as well as an acceleration of the same process under hypoxic conditions (*Kersten et al., 2022*).

In summary, macrophages and T cells interact in tumor immunometabolism and jointly regulate immune metabolism to influence tumor growth and immune surveillance. However, macrophage and T cell metabolism are increasingly seen as a complex, intricately regulated phenomenon that is affected by and can affect various features of tumor cells and TMEs. Therefore, future research and therapeutic efforts should not solely focus on macrophage and T cell metabolism but also take into consideration the intricate regulation of immune cell metabolism by both internal and external factors.

In this review, we describe recent advances in the three major metabolisms of macrophages and T cells in the context of cancer, summarize the altered and complex regulation of metabolic pathways that lead to macrophage and T cell polarization, and highlight key metabolites that regulate macrophage and T cell polarization and function. Finally, we discuss recent advances in reprogramming macrophage and T cell metabolism to generate effective antitumor responses.

## SURVEY METHODOLOGY

The PubMed online database was searched using three sets of subject terms: (1) macrophages, metabolic reprogramming, and immunotherapy; (2) T cells, metabolic

reprogramming, and immunotherapy; and (3) macrophages and T cells. Search criteria included a time span of the last 10 years, CAS 1 journals, and any type of literature. A total of 416 documents were retrieved and screened for relevance of subject terms, credibility of research methodology and validity of research content, and 142 documents were screened to exclude those with poor quality and duplicate content. Hua Cheng and Yongbin Zheng carefully analyzed and summarized the extracted key information, followed by a systematic introduction and in-depth discussion.

## Characterization of immune cells in the TME

TME consists of a variety of components, including tumor cells, tumor-associated fibroblasts, immune cells, extracellular matrix, and aberrant blood vessels. The morphology and function of various cells are different from those of normal tissues (Fig. 1) (Xia et al., 2021). In this complex, dynamic, and uncontrollable microenvironment, the functions performed by various types of non-tumor cells are also complex and dynamic. For example, immune cells among them will differentiate into different phenotypes, metabolic characteristics, and functional subpopulations, which play anti-tumor or pro-tumor roles, respectively, and further change TME through their metabolism, forming a complex and sophisticated network of interactions (Table S1) (Hinshaw & Shevde, 2019).

Activation of immune cells is the result of a collection of gene expression and environmental signals following antigenic stimulation. Whether they are innate or adaptive immune cells, they adjust their metabolic pathways in response to stimuli to enhance energy production and biomass synthesis, and alter the use of different metabolic pathways to promote proliferation, effector molecule production, and differentiation. In addition, when energy or biosynthetic pathways are disrupted, caution is required when interpreting complex data because manipulation of a single metabolic pathway may lead to impairment of other metabolic and signaling pathways (Buck, O'Sullivan & Pearce, 2015; O'Neill & Pearce, 2016). The phagocytic activity of macrophages removes dead and dying cells, and in turn, the removal of cellular debris provides nutrients to macrophages. However, tumors reprogram macrophage metabolism to prevent macrophage-mediated inflammation and kill tumor cells. The metabolic pathway of macrophages shifts and tends to polarize to the M2 type, releasing factors such as IL-10 and TGF-β, which inhibit T-cell activity. In contrast, the role of T cells in tumor immune metabolism is mainly in their metabolic regulation. T cells require large amounts of nutrients and oxygen for their proliferation and function, and they also release metabolites. In TME, the metabolic environment of T cells is very poor due to hypoxia, malnutrition, and metabolic toxicants, which can lead to a decrease in T cell activity. In addition, the metabolic regulation of T cells also affects their effects on tumor immunity. For example, T cell activity can be increased by increasing glucose transport and glycolytic metabolism of T cells, while memory responses of T cells can be promoted by inhibiting lipid metabolism (Kok, Masopust & Schumacher, 2022). These metabolic alterations not only directly affect immune cell function but also promote altered transcriptional activation status of functionally critical cytokines.

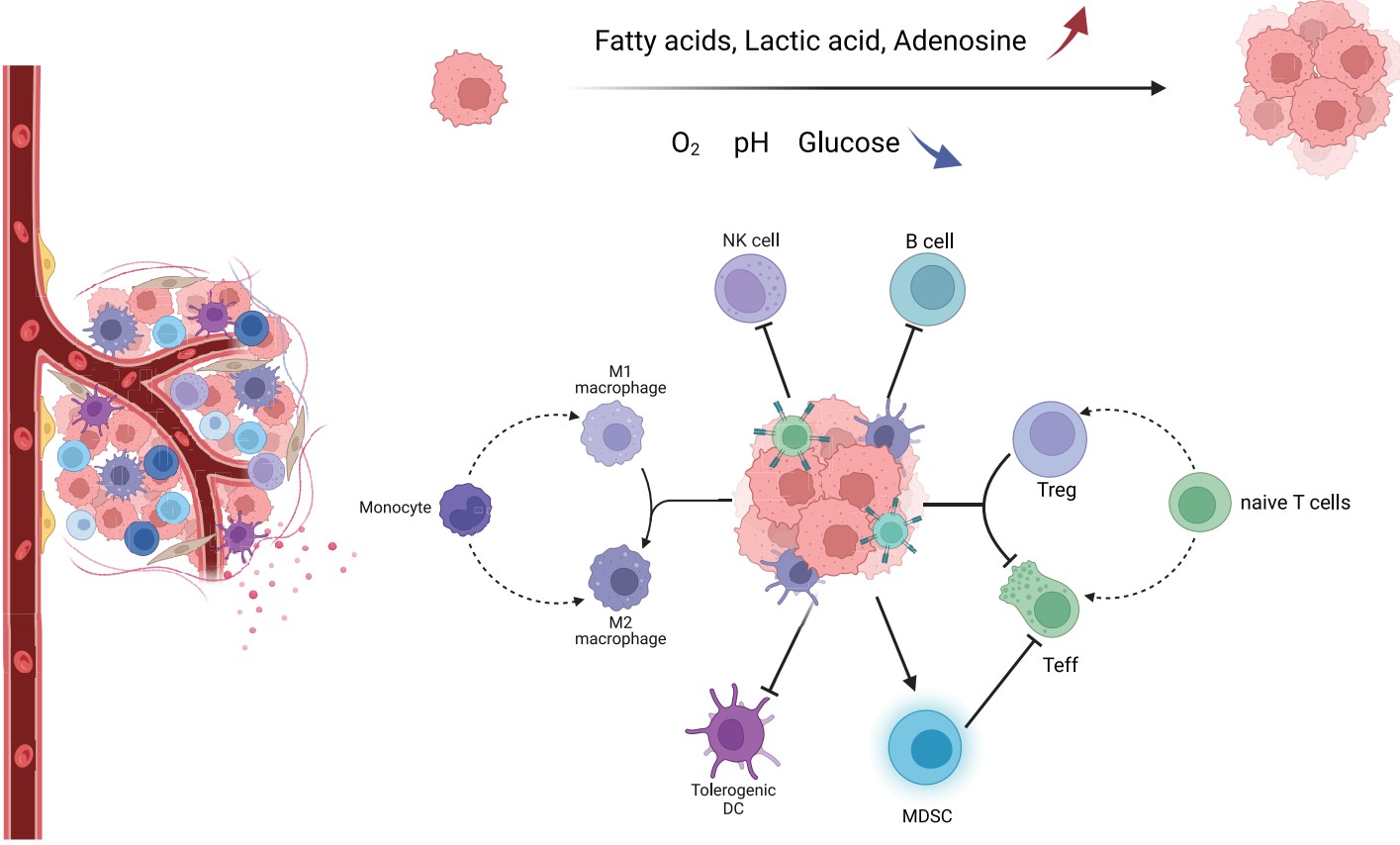

**Figure 1** **Major immune cells and metabolic changes in the TME.** TME mainly includes tumor cells, immune cells, tumor-associated fibroblasts, and aberrant blood vessels. As the tumor proliferates and metastasizes, TME gradually transforms into an immunosuppressive environment with low oxygen, low pH, low glucose concentration, high fatty acid concentration, high adenosine concentration, and high lactate concentration. Macrophages tend to have an M2 phenotype, DC cells have decreased antigen cross-presentation ability, MDSC has increased immunosuppressive ability, NK cells have decreased killing activity, B cells have decreased antibody secretion, and naive T cells are more activated as immunosuppressive Tregs.                                                                                               

## Macrophages

Macrophages are an important part of TMEs. In some solid tumors, macrophages make up more than 50% of the mass of TMEs with high plasticity and diversity (*Vitale et al., 2019*). Macrophage infiltration in solid tumors is associated with poor prognosis and chemotherapy resistance. Macrophages are essential for tissue homeostasis, but tumors distort this trend by stimulating macrophage proliferation, angiogenesis, and metastasis. In mouse models of cancer such as breast, pancreatic and endometrial cancer, macrophages promote tumor cell invasion, infiltration, and migration, facilitating cancer development and progression by suppressing anti-tumor immunity and stimulating angiogenesis (*Cassetta & Pollard, 2023*). At the site of metastasis, macrophages then promote tumor cell extravasation, survival and growth (*Hanahan & Coussens, 2012*; *Kitamura, Qian & Pollard, 2015*; *Noy & Pollard, 2014*; *Quail & Joyce, 2013*). These pro-tumorigenic activities are promoted by macrophage subpopulations that express typical markers but have unique transcriptional profiles. Mapping the TME through

single-cell RNA sequencing and mass spectrometry flow analysis of tumor and paracancerous tissues can identify macrophage subpopulations that are highly correlated with tumors, as well as associated specific markers, which will provide important targets for clinical treatment of the disease (*Obradovic et al., 2021*).

During the growth of a tumor, different metabolites released by malignant tumor cells and other TME cells can cause TAMs to change their energy metabolism. This changes the TAMs' structure and how they work. M1-type TAMs have a high flux of Aerobic glycolysis that produces reactive oxygen species (ROS) to kill tumor cells (*Mazzone, Menga & Castegna, 2018*). Type M2 TAMs are dependent on a high flux of OXPHOS and can generate vascular endothelial growth factor (VEGF) and IL-10 to enhance malignant cell growth (*Andrejeva & Rathmell, 2017*). However, the classification of macrophages is not a simple binary differentiation of M1 and M2; in fact, the division between them is blurred, and more transitional subtypes exist (*Boutilier & Elsawa, 2021*). Macrophages are subdivided into many different subtypes based on surface molecules, secretions, and functions, which are collectively referred to as "M1-like TAMs" and "M2-like TAMs", which more precisely describe their characteristics (*Mehla & Singh, 2019*). Both types of macrophages exist in different proportions at different stages of the tumor, and numerous studies have confirmed that more macrophages exhibit pro-inflammatory and anti-cancer type M1 in the early stage of the tumor and then transition to anti-inflammatory and pro-cancer type M2 macrophages in the process of tumor proliferation and metastasis (Fig. 2) (*Franklin et al., 2014*).

## M1-like TAMs

In the early stages of tumor formation, pro-inflammatory factors such as IL-12, Tumor necrosis factor-$\alpha$ (TNF-$\alpha$), Interferon-$\gamma$ (IFN-$\gamma$), bacterial lipopolysaccharide (LPS), Toll-like receptor (TLR) agonists, and pathogen-associated molecular patterns (PAMPs) in TME can lead macrophages toward an M1-like phenotype. The main function of M1-like macrophages is to secrete large amounts of inflammatory factors (*e.g.*, ROS, IL-6, TNF-$\alpha$, and NO) to kill tumor cells (*Li, Wan & Zhu, 2017*). Among them, ROS is crucial for macrophage phagocytic activity and the submission of antigens to T cells. To maintain their effective pro-inflammatory function and compete with tumor cells for nutrients, M1-like macrophages exhibit a tumor-like metabolic profile, also known as the "Warburg-like effect" (*Ghashghaeinia et al., 2019*).

M1-like macrophages ingest large amounts of glucose for glycolysis, intending to upregulate anabolism and provide a substrate for rapid energy supply, but at the cost of reduced glucose utilization efficiency and increased glucose consumption. At the same time, glycolysis upregulates pentose phosphate pathway (PPP) and synthesizes reduced nicotinamide adenine dinucleotide phosphate (NADPH) to protect itself from ROS damage. *In vitro* system, suppression of the glycolytic pathway in M1-like macrophages restored them to an undifferentiated state and reverted to the M1-like phenotype upon removal of the inhibitor, further confirming the central role of the glycolytic pathway in the regulation of macrophage polarization (*Suzuki et al., 2016*). To maintain phagocytic activity, mitochondrial metabolism of M1-like macrophages shifts from ATP synthesis to

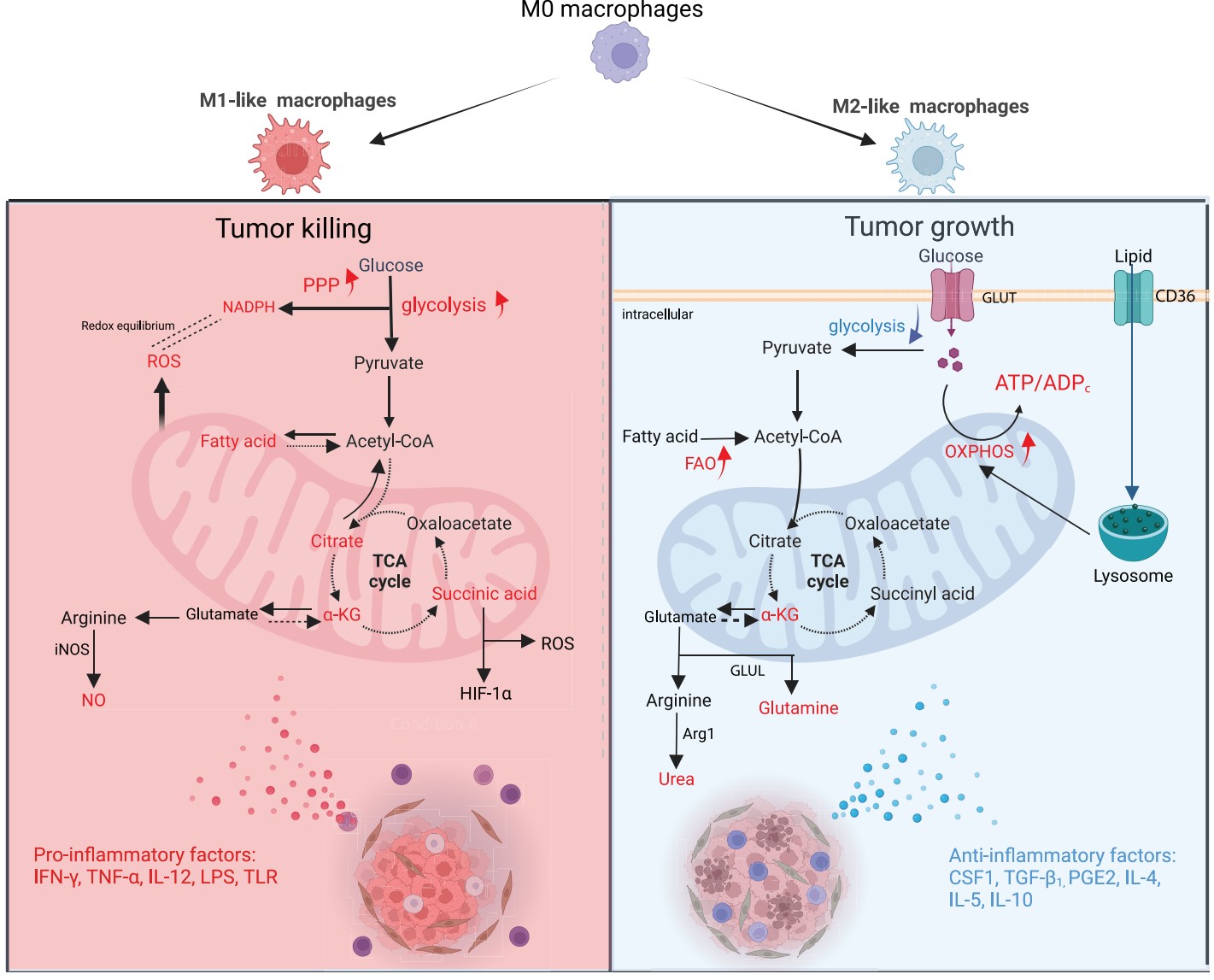

**Figure 2 Metabolic changes and phenotypic functions of macrophages in the TME.** In the early stage of the tumor, proinflammatory factors cause macrophages to polarise towards M1, which is mainly fed by glycolysis, PPP is upregulated and FAO is downregulated. The TCA cycle is broken in many places, resulting in metabolic accumulation. Mitochondria synthesize ROS in large quantities and arginine produces NO as a way to eradicate tumor cells. During the advanced stages of tumor development, anti-inflammatory factors cause macrophages to polarise towards M2, which is mainly fed by OXPHOS and has reduced glycolytic activity. The TCA cycle is mainly dependent on Gln uptake and arginine metabolism to urea, promoting tumor proliferation and metastasis. GLUT, glucose transporter protein; GLUL, glutamine ligase; α-KG; alpha ketoglutarate. Red text indicates increased metabolites.

ROS synthesis, and key enzymes in the tricarboxylic acid cycle (pyruvate dehydrogenase, isocitrate dehydrogenase, succinate dehydrogenase, *etc.*,) are inhibited and interrupted in several places, resulting in metabolite accumulation (citric acid and, succinic acid, *etc.*,) (*Jha et al., 2015*). Succinate promotes glycolysis by inhibiting proline hydroxylase activity and stabilizing hypoxia-inducible factor-1α (HIF-1α), it also enhances ROS production (*Liu et al., 2017b*).

In response to the need for cellular inflammatory factor biosynthesis (*Mazzone, Menga & Castegna, 2018*), M1-like macrophages turn down the fatty acid oxidation pathway and boost the *ab initio* synthesis of fatty acids. After LPS activates TLR4, macrophages turn up the *ab initio* fatty acid synthesis pathway. Glucose is burned to make citric acid, which is then turned into acetyl coenzyme A. Acetyl coenzyme A and NADPH (from PPP, which is also turned up) are substrates for fatty acid synthesis.

In terms of amino acid metabolism, the arginine metabolic pathway was most significantly altered. In M1-like macrophages, arginine is mainly metabolized by nitric oxide synthase to produce NO and citrulline, of which NO has an important effect on killing tumor cells (*Zhu et al., 2015*).

## M2-like TAMs

During tumor proliferation and metastasis, IL-4, IL-5, IL-10, IL-13, colony-stimulating factor 1 (CSF1), TGF-β1, and prostaglandin E2 (PGE2) in TME induce macrophages to polarize toward the M2-like phenotype (*Huber et al., 2016*). One of the important mechanisms by which M2-like macrophages promote tumor development is through the secretion of VEGFA, adrenomedullin (ADM), CXC chemokine ligand 8 (CXCL8), and CXCL12, which recruit and activate vascular epithelial cells and provide nutrients for their growth, thus promoting tumor tissue revascularization (*Biswas, Allavena & Mantovani, 2013*; *Hughes et al., 2015*). However, the resulting structural and functional abnormalities of the neovascular network cause the TME to exhibit varying degrees of hypoxia. On the one hand, the hypoxic state of the TME leads to an upregulation of lipid transport protein 2 (LCN2) and TAMs solute carrier family 40 member A1 (SLC40A1) expression, inhibiting the uptake of iron ions by TAMs, leading to iron ion accumulation in the TME, enhancing iron ion uptake by malignant tumor cells, further enhancing oxidative stress, activating signaling pathways such as Acute-Phase Response Factor 3 (STAT3) and nuclear factor kappa-B (NF-κB), and ultimately promoting tumor growth (*Mertens et al., 2016*; *Ören et al., 2016*). On the other hand, hypoxia leads to upregulation of DNA damage-inducible transcription factor 4 (DDIT4) expression, an endogenous inhibitor of its rapamycin target protein complex 1 (MTORC1) (*Wenes et al., 2016*), which inhibits glucose uptake by TAMs, leading to an increase in glucose content in TME, further increasing glucose utilization by tumor cells, and ultimately promoting tumor neovascularization and tumor metastasis.

Even under normoxia, succinate dehydrogenase and glyceraldehyde-3-phosphate dehydrogenase (GAPDH) activities were lower in TAMs than in normal macrophages, suggesting that M2-like TAMs also require a certain level of glycolysis to provide substrate and a rapid energy supply for cytokine synthesis, but in contrast, their glycolysis level was significantly lower than that of M1-like TAMs. Phosphorylation pathway for energy supply, with a high number of internal mitochondria and increased oxygen consumption rate, and its tricarboxylic acid cycle is highly dependent on glutamine (Gln) uptake. In colorectal cancer TMEs, GAPDH activity in M2-like macrophages was reported to be much lower than its activity in M1-like macrophages (*Miller et al., 2017*). The results of these experiments show that the decrease in glycolytic activity facilitates tumor uptake of

nutrients and promotes tumor development. However, in patients with medullary thyroid carcinoma, lactate production by tumor cells leads to a shift in the glycolytic pattern of M2-like TAMs from OXPHOS to glycolysis, secreting more lactate, TNF-α, and IL-6, which promotes tumor development (*Arts et al., 2016*). It has been reported in the literature that certain localized regions of M2-like TAMs have high glycolytic activity and could produce relatively high levels of lactate (*Donnem et al., 2018*). Taken together, the analysis suggests that despite relatively high levels of glycolysis of M2-like TAMs in certain parts of the TME, they still have a promotional effect on tumor growth (*Chen et al., 2019*).

M2-like macrophages must take in lipids through CD36 and then break them down with lysosomal acidic lipase in order to increase their OXPHOS levels, improve their residual respiratory capacity, live longer, and keep their M2 phenotype (*Mazzone, Menga & Castegna, 2018*). Even though FAO levels are higher in M2-like TAMs, some TAMs still add extracellular lipids to keep their metabolism going and play a role in modulating the immune system (*Hao et al., 2018*; *Xiang et al., 2018*). In this context, the expression of various proteins involved in lipid metabolism is abnormally mediated in these TAMs, such as fatty acid binding protein (FABP), monoglyceride lipase, medium-chain acyl-coenzyme A dehydrogenase, and autolytic enzyme domain five protein. In the initial phase of tumor progression, TAMs highly express FABP5, resulting in increased secretion of type I interferon (IFN-1) by TAMs to promote an anti-tumor immune response. In contrast, when tumors develop in advanced phases, TAMs highly express FABP4, which promotes the STAT3/IL-6 signaling pathway, thus facilitating tumor development. However, whether TAMs can promote tumor development by providing lipids directly to tumor cells like adipocytes remains to be proven (*Niu et al., 2017*). In addition, *in vivo* experiments showed that inhibition of FAO caused TAMs to polarize from M2 to M1 types in mouse models of lung and colon cancer (*Liu et al., 2017b*). In contrast, CSF1 from cancer cells leads to high expression of fatty acid synthase (FASN) in TAMs, and the resulting fatty acids activate peroxisome proliferator-activated receptor δ (PPAR-δ), which downstream releases the insulin-suppressing cytokine IL-10, thereby inducing TAMs to polarize to M2 type (*Hossain et al., 2015*; *Jha et al., 2015*).

In terms of amino acid metabolism, it is worth noting that M2-like macrophages use the opposite approach to their M1-like counterparts. Highly expressed Arg1 metabolizes arginine to ornithine and urea, promoting tumor cell proliferation, while highly expressed indole-2,3-dioxygenase (IDO) metabolizes tryptophan to kynurenine (KYN), inhibiting macrophage activation on T cells (*Zhu et al., 2015*). TAMs also have high levels of glutamine metabolism, and both amino acid transporter 2 (ASCT2) and glutaminase 1 (GLS1) were expressed at high levels in M2-like TAMs in a mouse malignancy model (*Choi et al., 2015*). *I. vitro* studies show that glutamine ligase (GLUL) promotes TAMs to M2 polarization by facilitating glutamate to glutamine transformation, and inhibition of glutamine uptake promotes TAMs to M1 polarization in mouse tumor models (*Liu et al., 2017b*; *Palmieri et al., 2017a*). The above studies suggest that there is a complex bidirectional metabolic regulation between tumor cells and TAM metabolism that is not only influenced by the immune response in TME but also counteracts the immune response.

## Metabolites and macrophage polarization

Lactate is the end product of aerobic glycolysis produced by tumor cell metabolism (*Goodwin et al., 2017*). Similar to aerobic glycolysis, lactate can directly regulate macrophage polarization (*Zhao et al., 2015*). Lactate stabilizes HIF-1α and induces VEGF expression and M2-type polarization of TAM, whereas under hypoxic conditions, the stability of HIF-1α and the induction of the oncogene signaling molecule MUC1 further promote the accumulation of lactate (*Shukla et al., 2017*). The activity of lactate is dependent on the expression level of HIF-1α and not on cytokines. Due to its small molecular size, lactate can penetrate TME more rapidly and deeply than large cytokine molecules. M2-type lactate polarization by macrophages is mediated by G protein-coupled receptor 132 (Gpr132), and deletion of Gpr132 in mice inhibits breast cancer metastasis (*Chen et al., 2017*). In addition, reduced expression of Gpr132 in breast cancer patients is associated with improved metastasis-free survival. It has been shown that secreted metabolites (*e.g.*, lactate) create a concentration gradient and transmit spatial information about tumor metabolism-related effects on macrophage polarization. This metabolite gradient also induces differential activation of signaling pathways in TAM, including the KRAS/MAPK pathway (*Carmona-Fontaine et al., 2017*). Thus, lactate levels in TME regulate the signaling function and polarization of M2 macrophages.

Adenosine is a very important part of how the immune system works. It controls the phagocytic activity of monocytes and macrophages through the G protein-coupled transmembrane receptors A1, A2A, A2B, and A3 (*Haskó & Pacher, 2012*). When adenosine binds to the A2A receptor, it stops macrophages from becoming M1-polarized during immune responses. On the other hand, when adenosine activates the A2B receptor, it makes macrophages become M2-polarized. Under culture conditions, adenosine promotes the migration and recruitment of monocytes to tumors. Additionally, TAM-expressed nucleic acid exonucleases CD39 and CD73 can catalyze adenosine production, signaling downstream adenosine receptors. Therefore, knockdown of the A2A receptor in myeloid cells inhibits melanoma growth and metastasis, leading to increased cytotoxic activity of Natural killer (NK) cells and T cells (*Cekic et al., 2014*). It also attenuates M2 polarization and enhances M1 polarization in TAM. Furthermore, hypoxia induces the expression of A2A and A2B receptors, reducing the expression of adenosine kinase and nucleoside transporter proteins to enhance adenosine levels in TME (*Morote-Garcia et al., 2008*). Therefore, the altered metabolism in TME promotes adenosine accumulation in the tissue interstitium, resulting in M2 polarization and immunosuppression.

Iron levels are also closely related to macrophages. M1 macrophages have been reported to have high ferritin expression and low expression of membrane iron transport proteins, whereas M2 macrophages have high expression of membrane iron transport proteins and low expression of ferritin (*Corna et al., 2010*; *Recalcati et al., 2010*). In addition to the direct provision of iron and iron-related proteins, macrophages may also influence iron metabolism in tumor cells through the release of cytokines, chemokines, and growth factors. Iron metabolism is also thought to contribute to macrophage polarization.

Iron-enriched diets induced hepatic and peritoneal macrophages in mice to polarize towards M2 and inhibited pro-inflammatory M1 phenotypes (*Agoro et al., 2018*). However, there are contrary reports that iron treatment of bone marrow-derived macrophages induced M1 marker expression and reduced IL-4-induced expression of M2 markers (*Zanganeh et al., 2016*). These contradictory findings illustrate the complexity of macrophage polarization in the context of iron metabolism. It was found that the expression level of xCT was upregulated in TAMs, and knockdown of xCT significantly reduced TAM infiltration, induced macrophage ferroptosis, and inhibited tumor growth and metastasis (*Tang et al., 2023*). It suggests that ferroptosis in TAMs plays an important role in regulating TME and anti-tumor immunity, providing new options for cancer treatment.

## Advances in immunotherapy with targeted TAMs

In the past few years, advancements have been made in the research of antitumor immunotherapies targeted at TAMs. These therapies aim to reprogram macrophages, counteract their immunosuppressive phenotypes, and reduce their numbers in the TME. Experiments have shown that CSF1R inhibitors can stop the transcriptional activation of CSF1R and its downstream signaling molecules in TAMs (*Ruffell & Coussens, 2015*). However, these inhibitors cause metabolic remodeling, which raises glycolysis levels in TMEs (*Liu et al., 2017a*). In the literature, it has also been said that lowering glycolysis levels and preventing lactate from building up in TME tumor cells may make the immunosuppressive effects of TAMs less strong. For example, Lactate dehydrogenase A (LDHA) is the critical enzyme for the glycolytic conversion of pyruvate to lactate, and inhibition of LDHA expression or activity can inhibit glycolysis in tumor cells, thereby inhibiting tumor development (*Brand et al., 2016*). In addition, the combination of the MTORC1-targeted activator phosphatidylinositol-3 kinase γ (PI3Kγ) with inhibitors of PI3Kδ mediates the downregulation of Pyruvate kinase M2 (PKM2) expression (*Locatelli et al., 2019*; *Weiss, 2020*). Blocking the expression of VEGFA also reverses M2-like TAMs to M1-like ones. This therapeutic strategy not only inhibits glycolysis but also tumor angiogenesis in TME (*Chen et al., 2019*; *Colegio et al., 2014*). Hypoxia suppresses anti-cancer immune sensitivity and can be reversed in the TME by increasing partial oxygen pressure. However, some blood vessels in cancer are abnormal and nonfunctional. Therefore, remodeling the vasculature in the TME is necessary to ameliorate hypoxia. Pro-oxidants can also be used to increase oxygen levels in the TME. In addition, inhibition of PD-L1 mediated by the hypoxia-inducible factor (HIF) pathway is an important strategy to enhance cancer immune responses. For example, inhibition of HIF-1α synthesis suppresses PD-L1 expression and induces lysosomal degradation of PD-L1, thereby enhancing the ability of CTL to kill cancer cells (*Huynh et al., 2023*). In addition, regulating iron metabolism, activating autophagy, or activating nitric oxide synthase 2 (NOS2), thereby promoting TAM metabolic reprogramming towards glycolysis and arginine catabolism, are effective strategies for targeting TAMs for malignancies (*Jayaprakash et al., 2018*; *Muliaditan et al., 2018*).

Because metabolites made by tumors affect how macrophages are reprogrammed, researchers are looking at lactate and adenosine as possible targets for cancer treatments. Lactate regulates macrophage metabolism through Gpr132, a highly expressed pH sensor in macrophages that modulates macrophage function in acidic TMEs of breast cancer, resulting in a significant reduction in tumor weight and volume in mouse models of breast cancer (*Chen et al., 2017*). It is important to note that Gpr132 is mainly expressed in lymphocytes and may therefore be the best minimal target for toxicity. Similarly, a study also showed that the tumor-derived lactic acid mechanism may be M2 polarization through upregulation of the ERK/STAT3 pathway (*Mu et al., 2018*), based on which the use of ERK/STAT3 inhibitors could attenuate lactic acid-induced macrophage polarization. Extracellular adenosine is another key tumor metabolite that can alter macrophage phagocytosis and cytokine production by binding to different adenosine receptors (A1, A2A, A2B, and A3). Recent studies have shown that adenosine can also control VEGF production by macrophages (*Ernens et al., 2010*). Therefore, targeting adenosine or adenosine receptors to regulate macrophage function is an effective approach to cancer treatment.

Glutamine synthetase (GS) is another crucial enzyme that promotes macrophage polarization to the M2 phenotype and facilitates tumor metastasis by increasing glutamine levels, potentially for cancer therapy (*Palmieri et al., 2017b*). Methionine sulfoximine (MSO) inhibits GS and suppresses the secretion of pro-inflammatory cytokines by mouse macrophages, restoring the M2 polarization towards the M1 phenotype in IL10-treated macrophages. MSO diverts glucose instead of glutamine into the tricarboxylic acid cycle in IL-10-treated macrophages, upregulating the expression of pro-inflammatory cytokines such as TNF-α, NO, and the chemokines CXCL9 and CXCL10 while decreasing the expression of CD163. Furthermore, if the GS gene is deleted in macrophages, they repolarize into the M1 phenotype, promoting the accumulation of cytotoxic T cells in tumors and weakening metastasis in the Lewis lung cancer model (*Palmieri et al., 2017b*).

Overall, metabolic changes in macrophages can affect their survival, polarization, and recruitment. Direct changes in metabolic flux or metabolite-controlled transcription factors are responsible for at least some of these responses. Secretion of metabolites such as adenosine and lactate can also fine-tune the response of macrophages in the TME to tumor cells. It should be noted that tumor cells and other cells in the microenvironment may have opposing biological functions when using the same shared pathways, depending on the cell's environment. For example, MTORC1 inhibitors that stop malignant cell glycolysis also stop TAM glycolysis and help grow new blood vessels in the tumor, which makes the tumor grow (*Wenes et al., 2016*). Restoring glycolytic levels of M2-like TAM is the basis for inducing their repolarization to M1-like types, but this strategy directly leads to competition between M2-like TAMs and other immune effector cells for limited glucose supply in the TME after restoring glycolytic levels, ultimately leading to immunosuppressive effects (*Weiss, 2020*).

In summary, although the tumor immunotherapy strategy targeting TAMs has made great progress and has vast potential in the future, there are still some issues that need to be addressed: (1) The molecular mechanisms regulating TAMs' metabolism and rapid

functional changes and their role in tumor progression; (2) the process of different TAM subgroup formation within the same TME; (3) the specific mechanism of TAMs' phenotypic changes before and after infiltrating tumors during cancer development; (4) the changes in TAMs' metabolism and immune response during immunotherapy; (5) the effects of other tumor-or stroma-secreted metabolites on macrophage polarization; (6) to what extent macrophage metabolic changes drive tumor cell evolution; (7) whether this reaction differs between primary and metastatic tumors. Addressing these issues will provide new insights for drug development targeting TAMs' immune metabolism and open up new pathways for finding effective tumor treatments. Although the treatment potential of cancer patients still requires further research, this study clearly shows that metabolic reprogramming is the main driving force for macrophage suppression or activation in TME and has the potential for immune-reactive cancer therapy.

## T cell

T cells are an essential component of the body's immune system that specifically targets and eliminates tumors. In the TME, T cells undergo complex interactions with tumor antigens and other cells, leading to a complex and diverse range of responses that ultimately determine the effectiveness of anti-tumor immunity. The fact that CD8[+] and CD4[+] T cells are mostly responsible for coordinating the immune response against tumors shows how important this interaction is *Chandran et al. (2015)*, *Lu et al. (2014)*. T cells that receive antigen stimulation must maintain a balance between basic nutrient supply and energy requirements during multiplication and differentiation. T cell subpopulations employ corresponding metabolic pathways such as glycolysis, OXPHOS, and FAO to adapt to TME nutrient levels and energy requirements. In turn, these metabolic pathways can control T cell activation and effector functions and also determine the direction of T cell differentiation (Fig. 3) (*Slack, Wang & Wang, 2015*). Depending on the surface antigen, initial CD4[+] T cells can differentiate into Th and Treg, whereas initial CD8[+] T cells differentiate into cytotoxic T lymphocytes (CTL). Different subpopulations of Th mediate immune functions, such as CTL and macrophages, by secreting various cytokines; Treg reduces the immune response by direct contact inhibition or by secreting suppressive cytokines; CTL mainly kills target cells by secreting substances such as perforin and granzyme B (GZMB), expressing apoptosis-related factor ligands, secreting TNF-α to bind to target cell surface receptors, and causing apoptosis in target cells.

Naive or resting T cells mainly depend on the OXPHOS pathway for ATP production. Once T cells are activated, the metabolic mode changes to glycolysis, glutamine, and branched-chain amino acid breakdown metabolism. This causes more glucose and amino acids to be taken in *Wang et al. (2011)*. Activated T cells exhibit a paradoxical phenomenon whereby they increase fatty acid uptake while simultaneously inhibiting the fatty acid oxidation process and promoting lipid synthesis through mitochondrial oxidative phosphorylation augmentation (*O'Sullivan et al., 2018*). In addition to enhanced glycolysis, the pentose phosphate pathway also enhances glucose metabolism, which, along with glutamine catabolism, promotes basic biomolecular anabolism (*Chang et al., 2013*; *Swamy et al., 2016*). These metabolic changes are mediated by T cell receptors and CD28, as well as

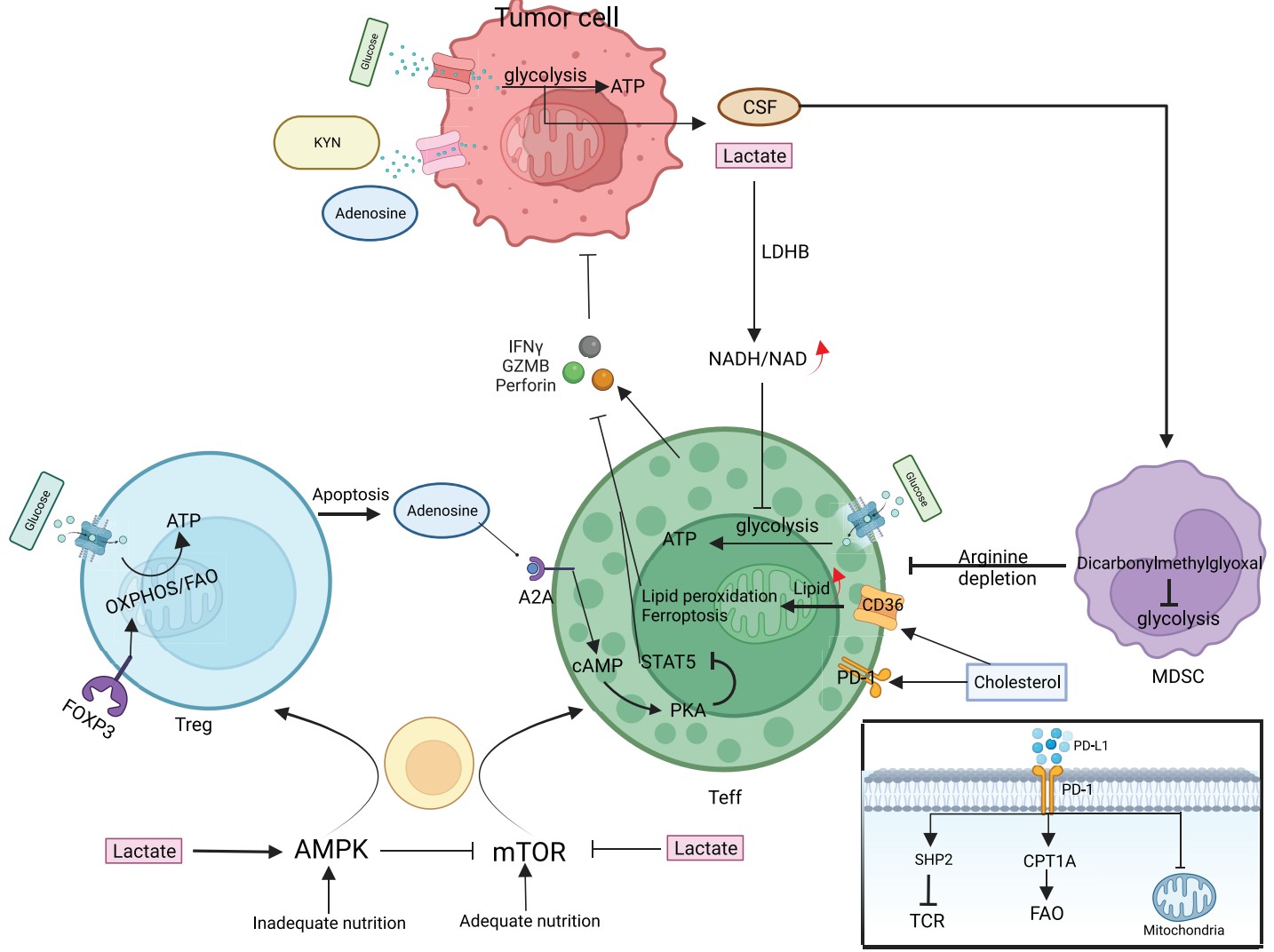

**Figure 3 Metabolic regulation of T cells and tumor cells in the TME.** mTOR/AMPK determines the fate of T cell differentiation. Teff is mainly glycolytically fed, secreting perforin, granzyme, and interferon γ to kill cancer cells. Treg attenuates teff activity by direct contact or secreting adenosine. Tumor cells also use glycolysis for energy supply, deplete nutrients in the microenvironment, and secrete large amounts of lactate, adenosine, and KYN to inhibit Teff. MDSC proliferation activation selectively depletes arginine and suppresses the immune function of T cells. Cholesterol in the microenvironment upregulates CD36 and PD-1 expression and CD36 mediates fatty acid uptake and induces Teff lipid peroxidation and ferroptosis. The PD-1/PD-L1 signaling pathway can attract the SHP-2 protein, which is responsible for regulating TCR signaling-induced T cell activation. Simultaneously, increasing the expression of CPT1A can augment FAO, which effectively attenuates T cell activation, and can also cause changes in mitochondrial ultrastructure, leading to mitochondrial dysfunction. CSF, Colony-stimulating factor; LDHB, lactate dehydrogenase B; GZMB, granzyme B; SHP2, Src homology-2 domain-containing protein tyrosine phosphatase; CPT1A, carnitine palmitoyl-transferase 1A.

downstream signaling pathways activated by cytokine receptors such as the mammalian target of rapamycin (mTOR)/protein kinase B (Akt)/PI3K regulation. mTOR is a key modulator of T cell differentiation, consisting of mTORC1 and mTORC2, and coordinates cellular variation in response to nutrients and energy (*Delgoffe et al., 2011*). mTOR induces C-Myc and HIF-1α expression, both of which negatively regulate the mTOR complex. C-Myc carries out lipid, amino acid, and nucleic acid synthesis by promoting aerobic

glycolysis and the expression of glutaminolysis metabolizing enzymes (*Wang et al., 2011*). HIF-1α-mediated T-cell reaction to oxygen also enhances glucose intake and catabolism and synthesizes cytokines secreted by activated Teff, facilitating effective activation and clonal proliferation of Teff (*Cho et al., 2019*). It also inhibits Treg differentiation and function by regulating lipid metabolism gene expression (*Angelin et al., 2017*). When the antigen is cleared, a portion of Teff differentiates into persistent Tm, and the metabolic pattern changes from glycolysis to OXPHOS-mediated catabolism (*Pearce et al., 2009*).

## Inhibition of antitumor Teff

Due to the high glycolytic capacity of tumor cells and poor exchange of vascular nutrients, TME is often glucose-depleted or hypoglycemic, putting pressure on glucose competition between cells. Adenosine monophosphate-activated protein kinase (AMPK) and mTOR, respectively, control the building up and breaking down of T cells (*Inoki, Kim & Guan, 2012*). When nutrients and energy are sufficient, mTOR is activated and induces glycolysis-based anabolic reactions (*Zeng et al., 2013*). When there aren't enough nutrients and energy, the AMPK pathway turns on and turns off mTOR. This causes a change in cellular metabolism from anabolism to catabolism based on mitochondrial OXPHOS and FAO, but it's not clear how mTOR and AMPK work together (*Blagih et al., 2015*). Both Teff cells and tumor cells have a high glycolytic capacity, which has a number of bad effects: it stops T-cells from activating and making effectors, which makes them more vulnerable to apoptosis, which in turn makes T-cells fail and, in some cases, lets tumors escape the immune system. In addition, rapid tumor cell glycolysis may indirectly inhibit Teff function by affecting other immune cells, *e.g.*, enhanced aerobic glycolysis of tumor cells increased levels of granulocyte-macrophage colony-stimulating factor (GM-CSF) and granulocyte colony-stimulating factor (G-CSF), promoted the multiplication of MDSC, and further inhibited T cell activation. Intermediates in T cell glycolysis may have a regulatory role in immune function, and phosphoenolpyruvate (PEP), a glycolytic intermediate, has been reported to maintain the $Ca^{2+}$-activated T cell receptor-mediated nuclear factor signaling pathway. Overexpression of phosphoenolpyruvate carboxykinase1 in cells allows tumor-specific reprogramming of $CD4^+$ and $CD8^+$ T cell metabolism, increases PEP production, and enhances Teff killing through cycloisomerization, and also shows results in limiting melanoma growth and prolonging survival time *in vivo* experiments in mice (*Ho et al., 2015*).

The Warburg effect causes cancer cells to consume glucose and increase lactate, which accumulates in large amounts and leads to a pH of 5.85 in TME, which is 40 times higher than normal (*Watson et al., 2021*). Lactate inhibits the PI3K/Akt/mTOR pathway, thereby suppressing T cell multiplication and IFN-γ generation and reducing the ability of Teff to kill tumors (*Düvel et al., 2010*). Lactate can also produce pyruvate and reduced nicotinamide adenine dinucleotide (NADH) using LDHB, resulting in an imbalance in pyruvate production and a lower nicotinamide adenine dinucleotide (NAD)/NADH ratio, which impairs T cell multiplication and killing by blocking aerobic glycolysis (*Quinn et al., 2020*). In addition, it has been shown that the V-domain Ig suppressor of T-cell activation (VISTA) selectively binds P-selectin glycoprotein ligand 1 under weakly acidic conditions

at pH 6.0 and participates in the suppression of T-cell function. *In vivo* experiments in mice showed that the VISTA acid pH-selective blocking antibody showed advantages in targeting and persistence compared to non-pH-selective antibodies. This suggests that TME pH may also influence the immune response and may serve as an entry point for therapy (*Johnston et al., 2019*). Considering the intracellular lactate content, targeting monocarboxylate transporter protein 1/4, which transports lactate, could lead to both a decrease in lactate accumulation and glycolysis levels in tumor cells, causing tumor cell death and avoiding lactate inhibition of T cells, promoting secretion of IFN-γ from T cells. However, given that LDHB limits T-cell proliferation, treatment should be considered in terms of regulating lactate levels, and the need for lactate for antitumor immunity should not be overlooked.

A common feature of altered metabolism in TME is lipid accumulation. The accumulation of lipid droplets in immune cells and enhanced fatty acid oxidation often tend to manifest as immunosuppression. Microenvironmental cholesterol causes CD8[+] T cells to exhibit PD-1, which results in T cell depletion in tumors (*Ma et al., 2019*). CD8[+] T cells upregulate CD36 expression, which mediates fatty acid uptake and induces lipid peroxidation and ferroptosis, reducing their production of toxic cytokines. Therefore, targeting CD36 effectively restores CD8[+] T cells' anti-tumor activity, and overexpression of glutathione peroxidase 4 eliminates lipid peroxidation and restores CD8[+] T function (*Ma et al., 2021*; *Xu et al., 2021*). PGE2 is a small-molecule lipid mediator synthesized by arachidonic acid *via* cyclooxygenase 2 (COX 2). PGE2 inhibits T-cell mobilization and Th1 differentiation by suppressing IL-2 production and downregulating transferrin receptor levels by the cyclic adenosine monophosphate (cAMP) signaling pathway (*Kalinski, 2012*; *Kunikata et al., 2005*; *Wehbi & Taskén, 2016*).

In terms of Gln metabolism, Teff is like tumor cells in that activated Teff needs more Gln metabolism to make the intermediates it needs to grow, but there are still differences between the two in how flexible their metabolisms are (*Leone et al., 2019*). In recent years, other types of amino acids have been identified in T cell immune function, such as arginine, which is vital for T cell survival and antitumor activity. Higher arginine concentrations favor large T cell proliferation, tend to differentiate initial CD4[+] cells into memory T cells rather than Th cells, and are beneficial for CD4[+] and CD8[+] Teff survival (*Geiger et al., 2016*). Glycine and single-carbon units produced by serine metabolism are available for nucleotide synthesis and are important for Teff cell expansion. The high rate of tumor cell metabolism depletes amino acids in TME, which may be a way for tumors to reduce T cell function and evade immune responses.

Specific metabolites of tumor cells also inhibit Teff activity. Tryptophan turns on and boosts T cells. Tumor cells use IDO and tryptophan-2,3-dioxygenase (TDO) to turn tryptophan into KYN, which stops Teff activity (*Liu et al., 2009*; *Zhai et al., 2018*). KYN binds to the aryl hydrocarbon receptor, which turns on transcription factors in the cytoplasm and causes CD4[+] T cells to change into Tregs (*Quintana et al., 2008*; *Sinclair et al., 2018*). KYN also makes tumor-infiltrating lymphocytes make ROS, stops IL-2 signaling, and stops Tm from working (*Dagenais-Lussier et al., 2016*). Adenosine, a metabolite that CD38, CD39, and CD73 produce when they are enzymatically active, has

immunosuppressive effects (*Vijayan et al., 2017*). Adenosine produces cAMP by binding to the A2AR; cAMP activates protein kinase A (PKA), and PKA mediates Akt inhibition of the signal transducer and STAT5 phosphorylation, which inhibits T cell function (*Bono et al., 2015*; *Zhang et al., 2004*). Furthermore, PKA phosphorylates cAMP-responsive binding protein (CREB) and induces Treg production (*Wen, Sakamoto & Miller, 2010*). Because tumor cells and macrophages express CD38, CD39, and CD73, TME is enriched in adenosine, which maintains its suppressive nature.

The metabolites or suppressive factors of other immunosuppressive cells in TME are also important for the effects of T cells. For example, arginine is an essential amino acid in protein synthesis, and increasing its level can lead to a complete metabolic change from glycolysis to OXPHOS, thus sustaining Tm survival (*Geiger et al., 2016*). In TME, amino urea-sensitive ammonia oxidase in MDSC catalyzes the production of dicarbonyl methylglyoxal and its accumulation in cells, reducing the level of MDSC glycolysis, the metabolic dormancy characteristic of MDSC. MDSC binding to T cells triggers a cascade of events, leading to the entry of dicarbonyl methylglyoxal into cells and binding with L-arginine, resulting in the depletion of L-arginine and thereby inhibiting the immune function of T cells. Targeting arginase or L-arginine supplementation may improve the therapeutic effect (*Baumann et al., 2020*).

Other cell surface molecules in TME also play a regulatory role in the metabolism of important nutrients in T cells. PD-1 acts primarily by affecting T cell antigen receptor signaling and T cell co-stimulatory activation. PD-1/PD-L1 pairing recruits SHP-2 phosphatase to the intracellular region of PD-1 molecules, dephosphorylates the T cell receptor (TCR) and CD28 downstream signaling, and inhibits T cell activation mediated by the TCR signaling pathway (*Hui et al., 2017*). The PD-1 signaling pathway also influences the ultrastructure of mitochondria, thereby reducing the expression of mitochondrial cristae histones Mic19 and Mic14, resulting in reduced mitochondrial cristae production in T cells and weakened mitochondrial depolarization, which in turn leads to mitochondrial dysfunction (*Ogando et al., 2019*). PD-1/PD-L1 binding reprograms T cell metabolism and enhances FAO endogenous lipid rate-limiting enzyme. Expression of the carnitine palmitoyltransferase 1A gene (CPT1A) enhances FAO, impairs cell glycolysis, glutaminolysis, and branched-chain amino acid metabolism, and inhibits energy and substance synthesis required for T cell activation, but can rescue Teff from glycolysis-induced rapid death and terminal cellular differentiation, tilting the metabolic balance towards an adipose-based metabolic model and allowing T cell lifespan to be prolonged (*Patsoukis et al., 2015*). Thus, the blockade of PD-1 signaling initiates Teff glycolysis, leading to its terminal fractionation, clonal deletion through apoptosis, and consequently a decrease in Teff availability (*Chowdhury et al., 2018*). This is likely the reason why some patients initially respond but later do not respond during anti-PD-1 antibody therapy.

The hijacking of immune cell mitochondria by cancer cells is also rapidly attracting attention. A study in late 2021 using field-emission scanning electron microscopy (FESEM) observed for the first time that cancer cells are able to erect tunneling nanotubes (TNTs) to take the mitochondria of immune cells, leading to a heavy trauma to the

immune cells. The presence of TNT facilitates intercellular communication and organelles sharing, and also allows cancer cells to benefit from it. This is thought to be an immune escape strategy. However, the questions of how TNT is formed and how cancer cells rob T cell mitochondria have yet to be resolved.

## Enhancement of immunosuppressive Treg

Unlike Teff cells, Treg cells mostly depend on tricarboxylic acid cycle-coupled OXPHOS and FAO to stay alive and change (Ho et al., 2015). Oxidative phosphorylation in Tregs is stronger and more sensitive to the amount of free oxygen in the TME because the nuclear transcription factor red blood-related factor 2 is linked to a weaker antioxidant system. This makes them more likely to undergo apoptosis in response to oxidative stress. Treg apoptosis converts a large amount of ATP to adenosine and releases it, which binds to receptors on antigen-presenting cells, Teff, maintaining and amplifying its immunosuppressive capacity (Maj et al., 2017).

Treg refers to T cells that are positive for forkhead box P3 (Foxp3), CD25, and CD4. Foxp3 can be used not only as a characteristic marker to distinguish Treg but also as an important metabolic transcription factor. Foxp3 expression can lower levels of Myc and glycolysis, raise levels of electron transport chain complex expression, and improve oxidative phosphorylation and NAD oxidation. The low glucose environment in TME can induce Foxp3 expression and cause T cells to tend to differentiate into Tregs, aiding immune escape from tumors (Angelin et al., 2017). Therefore, when exploring new methods to inhibit tumor development, targeting the different oxidative metabolism characteristics of Treg and Teff may also be considered, as may avoiding impairing the normal function of other immune cells as much as possible.

Teff's dependence on different substances in TME had the opposite effect on Treg, like stopping the buildup of lactic acid or lipids in Teff, but it helped Treg live and work. Treg has a strong ability to oxidize exogenous lactic acid, thus improving its survival rate in TME. Mice with knocked-out lactate-related signaling pathways make less Treg and have lower levels of IL-10. This suggests that targeting lactate-related receptors or pathways could be a good way to treat a disease (Ho et al., 2015). CD36 transport lipid is also upregulated in Treg, but at the moment it works as a central metabolic regulator for Treg survival, promoting reprogramming and making Treg more adaptable to lactate-rich environments (Wang et al., 2020). Although tumor cells depend on Gln metabolism, Teff also requires Gln. If there is not any Gln in the environment, it will change how Th1 cells tend to develop and cause the first CD4$^+$ T cells to change into Treg cells, which help tumors grow. When the Gln transporter ASCT2 was taken away from T cells, it made it harder for Th1 and Th17 to work, but it did not affect Tregs. Although many problems need to be solved, in recent years there has been some progress in research on the different dependence of tumor cells, anti-tumor immune cells, or pro-tumor immune cells on Gln (Ho et al., 2015), and this perspective still has the potential for tumor treatment (Ho et al., 2015).

In summary, it is clear that Teff cells are metabolically similar to tumor cells, and Treg cells have a survival advantage in TME. So, the goal of tumor therapy is to find out more

about the key differences between the metabolisms of Teff and tumor cells, or to find metabolic pathways that regulate Teff in the opposite way as Treg. This will be the starting point for looking into new metabolic therapies to reduce the viability of tumor cells, stop Treg from suppressing the immune system, and make Teff more effective at killing tumor cells. This will help to explore new metabolic therapies and achieve the best anti-tumor effect.

## Advances in targeted T-cell immunotherapy research

There are two main types of tumor immunotherapy: cellular immunotherapy and immune checkpoint inhibitor therapy. Cellular immunotherapy refers to taking immune cells from the patient's body and transforming them outside the body so that these cells have a more effective and precise immune capacity against cancer cells. After the transformed immune cells are injected back into the patient's body, they will be targeted to destroy cancer cells. Currently, T cell-based cellular immunotherapies include CAR-T therapy, TCR-T therapy, TIL therapy, and CTL therapy (*Waldman, Fritz & Lenardo, 2020*). Among them, the most high-profile CAR-T therapy is chimeric antigen receptor T cell immunotherapy, in which T cells are genetically engineered to be activated and equipped with a localized CAR navigation device (tumor chimeric antigen receptor) to transform T cells, which are ordinary "warriors", into "super warriors". CAR-T cells are specifically designed to identify tumor cells in the body and efficiently kill them, thus achieving the goal of treating malignant tumors." CAR-T therapy is entering a blowout phase with approval of Aquilensai and Regiorense in China in 2021 and FDA approval of Cedar Key Orense in 2022. A large part of them have achieved good results. Although the clinical efficacy of CAR-T therapy in solid tumors is poor, CAR-T therapy has made excellent progress in gastric cancer, liver cancer, pancreatic cancer, and other areas in recent years. As the first international CAR-T cell targeting Claudin 18.2, CT041 achieved an objective remission rate of 48.6% and a disease control rate of 73% for all patients with digestive system tumors; an overall objective remission rate of 57.1% and a disease control rate of 75.0% for all gastric cancer patients, which is the largest clinical sample size study of CAR-T cells for solid tumors to date (*Qi et al., 2022*). Apart from the typical complications associated with cancer treatments, CAR-T cell therapy also has specific side effects that are not common with other treatments. These specific effects include cytokine release syndrome (CRS) and immune effector cell-associated neurotoxicity syndrome (ICANS), which are the two most common direct complications of CAR-T cell therapy. The severity of these two complications directly relates to the design of the CAR-T cell product, treatment regimen, and patient and tumor factors. Beyond their potential for treating cancer, CAR-T therapies also show promise for treating autoimmune diseases, chronic infections, fibrotic disorders, and even aging (*Baker et al., 2023*).

Immune cells produce small molecules of proteins that suppress themselves, and tumor cells use this mechanism to suppress immune cells and escape from the body's immune system to survive. Immune checkpoint inhibitors (ICI) lift this suppression, allowing immune cells to reactivate their work and destroy cancer cells. There is a close correlation between T-cell immune checkpoint inhibitors and T-cell immune metabolism. Immune
checkpoint inhibitors can affect the metabolic activity of T cells and thus regulate the immune response of T cells. These inhibitors can activate metabolic pathways that target cells, such as glycolysis and lipid metabolism, thereby enhancing T-cell growth and differentiation. At the same time, they can also inhibit metabolic pathways, such as oxidative phosphorylation pathways, thereby reducing cellular energy metabolism and function and affecting tumor infiltration and T cell killing capacity, thereby improving immunotherapeutic efficacy. In addition, T cell immunometabolism and T cell immune checkpoint inhibitors can interact with each other. The CD28 costimulatory molecule is an important T cell costimulatory molecule that enhances the T cell immune response by regulating glycolytic metabolic pathways. Certain immune checkpoint molecules can inhibit CD28 signaling, thereby reducing T cell metabolic activity. Therefore, the use of both immune checkpoint inhibitors and drugs that regulate metabolic pathways in therapy may be more effective in activating T-cell immune responses and improving therapeutic efficacy.

The use of ICI has increased significantly in recent years for the treatment of cancer. In 2013, ipilimumab, which specifically targets cytotoxic T lymphocyte-associated antigen-4 (CTLA-4), was approved for clinical use in cancer patients, and PD-1/PD-L1 monoclonal antibodies have shown significant efficacy in a lot of malignancies, including lung, liver, and colon cancers, making immunotherapy gradually a mature tool for cancer treatment (*Ansell et al., 2015*; *Borghaei et al., 2015*). However, it is clear from large clinical trials and treatments that only a minority of patients respond to immunotherapy, and there are still a large number of patients with no or low immune response and recurrence, making ICI combinations a recent research hotspot (Table S2) (*Llosa et al., 2019*).

The balance between mTOR and AMPK determines the destiny of T cells. Teff is dependent on the mTOR pathway, while Tm is more dependent on AMPK. Metformin for the treatment of type two diabetes has anticancer effects. Metformin increases pAMPK levels, decreases pS6 downstream mTOR protein levels, and prolongs Tm lifespan. However, most of these studies were suprapharmacologic dosing, using doses of metformin that exceeded the clinical dose by up to 10 times, and the effects of high concentrations of metformin in *in vitro* studies could not be easily replicated in preclinical or clinical studies (*Yu & Suissa, 2023*). mTOR inhibitor rapamycin enhances PD-L1 monoclonal antibody against oral cancer cell line MOC1 inhibition, amplifies tumor-infiltrating Tm, enhances IFN-γ secretion, and promotes the expression of MHC-I in tumor cells. Vistusetib (AZD 2014), a mTORC1/2 dual kinase inhibitor, promotes Th1 differentiation and enhances Tm function and lifespan. In combination with PD-1/PD-L1 monoclonal antibodies, it inhibits the functional depletion of tumor-infiltrating lymphocytes and prolongs survival in MC-38 and CT-26 transplanted tumor animals (*Eikawa et al., 2015*).

Teff enhances lipid biosynthesis, while Tm decreases lipid synthesis and enhances FAO. Benzofibrate is a PPAR-1α agonist that promotes CPT1α, lipoyl coenzyme A dehydrogenase (LCAD), and PPARγ co-activator-1α (PGC-1α) expression, enhances FAO and tumor infiltration ROS in lymphocytes, maintaining their function, and has significant inhibitory effects on lung cancer in combination with PD-L1 monoclonal

antibody (*Wan et al., 2020*). GW501516 is a PPARα and PPARδ/β agonist that enhances its CPT1α expression, promotes FAO, and promotes Teff differentiation in CD8$^+$ T cell overt immunotherapy, and in combination with PD-1 monoclonal antibody, has significant efficacy in animal models of melanoma (*Saibil et al., 2019*).

In a mouse model of PD-1 monoclonal antibody therapy, *Chamoto et al. (2017)* found that tumor-infiltrating CD8$^+$ T cells draining lymph nodes had more mitochondria and ROS. Teff and Tm's ROS were synergistically inhibited with PD-1 monoclonal antibodies on tumor cells. Carbonyl-cyano-p-trifluoromethoxyphenylhydrazine is a mitochondrial uncoupler that reduces the mitochondrial membrane potential of T cells, and Luperox is an H$_2$O$_2$ precursor, both of which promote ROS production, activate PGC-1α and downstream signaling, and enhance Teff function (*Chamoto et al., 2017*).

*Sukumar et al. (2013)* found that activated CD8$^+$ T cells made more Teff when the glycolysis inhibitor 2-deoxyglucose was added. *Leone et al. (2019)* pointed out that the transglutamine antagonist 6-diazo-5-oxo-L-norleucine inhibited tumor cell OXPHOS and glycolysis, increased TME oxygen content, decreased acidity, increased Teff OXPHOS metabolism, and promoted Teff differentiation. The combination of both with PD-1 monoclonal antibodies increased T-cell antitumor activity (*Leone et al., 2019*).

CB-1158 is an arginase inhibitor. *Steggerda et al. (2017)*. found that CB-1158 alleviated the suppression of T cell proliferative by MDSC *in vitro* and, in combination with ICI, increased the number of tumor-infiltrating natural killer (NK) cells and CD8$^+$ T cells in an *ex vivo* model of tumor cell lines CT26, B16, and 4T1. *He et al. (2017)* established a BALB/c mouse transplantation tumor model with *in situ* and metastatic osteosarcoma and found that L-arginine significantly increased the levels of splenic CD8$^+$ T cells and serum IFN-γ in mice, protected expanded CD8$^+$ T cells from depletion in combination with PD-L1 monoclonal antibody, and enhanced the capability of these T cells to secrete GZMB, IFN-γ and perforin.

BGB-5777 is an IDO1 inhibitor that makes Teff work better and decreases KYN production by blocking IDO1 and stopping the breakdown of tryptophan. Combination with a PD-1 monoclonal antibody consistently enhances survival benefits in patients with progressive glioblastoma (*Ladomersky et al., 2018*). PEG-KYNase is a drug-degrading enzyme that decomposes KYN into non-toxic, easily cleared, immunologically inert metabolites, reversing the immunosuppressive impacts of IDO1/TDO upper regulation and inhibiting tumor growth. The combination of PEG-KYNase with PD-L1 was effective in treating CT26 colon cancer, 4T1 breast cancer, and B16-F10 melanoma cancer in a mouse transplantation tumor model (*Triplett et al., 2018*).

Adenosine is produced by CD39 and CD73 active extracellular enzymes, and to block the adenosine pathway, *Perrot et al. (2019)* prepared two antibodies, IPH5201 and IPH5301, which target the surface of human cell membranes and soluble CD39 and CD73 and effectively block ATP hydrolysis to adenosine. These antibodies promote anti-tumor immunity by stimulating dendritic cells, macrophages, and T cells (*Perrot et al., 2019*). IPH5201 increases the antitumor activity of the ATP-induced chemotherapeutic agent oxaliplatin in a CD39 knock-in mouse model. CPI-444 and PBF509 are potent and selective A2AR antagonists, and blocking A2AR with both restores adenosine-induced

inhibition of signaling by T cells and promotes IL-2 and interferon-γ production. *I. vitro* studies found that CPI-444 combined with PD-L1 monoclonal antibody or CTLA-4 monoclonal antibody eliminated up to 90% of mouse tumors, including restoration of incomplete immune response to PD-L1 monoclonal antibody or CTLA-4 monoclonal antibody monotherapy, and tumor-healed mice were completely inhibited in growth after tumor reinoculation, indicating CPI-444 and PBF509 inhibit CD8$^+$ Tm immune deletion and prolong their lifespan (*Willingham et al., 2018*).

## CONCLUSION AND OUTLOOK

TME is a complex and changing microenvironment. When immune cells enter TME, they cause a series of metabolic changes that often slow down the immune system and help tumors grow and spread. Based on these traits, regulating the balance between tumor metabolism and immune metabolism may be a way to improve the function of immune cells inside the tumor. This could lead to a new direction for tumor immunotherapy, including a series of immunometabolic therapies such as inducing macrophage polarization, stopping macrophage recruitment, increasing key T cell glycolytic enzymes, and boosting lipid synthesis.

Of course, there are not only macrophages and T cells as described above in TME but also various other important immunocytes that perform different anti-tumor or pro-tumor functions, for instance, B cells, NK cells, dendritic cells, neutrophils, *etc.*, which are subject to complex metabolic regulation. Although immunometabolic therapies require a combination of effects on tumor cells themselves as well as on various cell subpopulations, which is challenging, research into such therapies is undoubtedly promising.

### Funding
The authors received no funding for this work.

### Competing Interests
The authors declare that they have no competing interests.

### Author Contributions
- Hua Cheng performed the experiments, analyzed the data, prepared figures and/or tables, authored or reviewed drafts of the article, and approved the final draft.
- Yongbin Zheng conceived and designed the experiments, authored or reviewed drafts of the article, and approved the final draft.

### Data Availability
This article is a literature review.

## Supplemental Information

Supplemental information for this article can be found online at http://dx.doi.org/10.7717/peerj.16825#supplemental-information.

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
