# Peer review of "Advances in macrophage and T cell metabolic reprogramming and immunotherapy in the tumor microenvironment"

_PeerJ, doi:10.7717/peerj.16825_

## Round 0.1 · original submission · Minor Revisions

Four experts assessed the manuscript and found the content relevant. Some suggestions for improvement should be straightforward and that without a doubt would strengthen the manuscript content.

·

Basic reporting

The work by Hua Cheng and Yongbin Zheng presents a review of the metabolic state of T cells and the effect of metabolic reprogramming in the tumor microenvironment.
This reviewer thinks the review is well-written, and the figures are clear and pertinent. I find this review easy to follow and with a topic that has taken so much relevance in the past few years.
In the following lines, I provide some comments regarding the content of the review that I encourage the authors to consider for expanding the review's aim. Also, none of the additional references I suggest are from my group or close colleagues.

Experimental design

The methodology is good; however, this reviewer thinks that eliminating papers from a determined impact factor may have overlooked a paper that may have relevant information. I kindly request to clarify this issue.
I suggest adding a table to summarize the cells, cell products, and metabolites involved in each stage of TME and their references to provide the reader with a compendium of the most relevant events. This will be a reference for researchers working in this field.

Validity of the findings

The review contains interesting aspects of the tumor microenvironment and points to complementary therapeutic approaches for enhancing immune response to tumors. In this reviewer's opinion, the authors may consider some aspects. Additionally, I suggest adding a table that describes the types of tumors associated with the aspects discussed in this review and the main differences in TME found to date.
One minor criticism this reviewer has is that in the introduction, authors are encouraged to define the metabolic aspects the review will seek to convey and the role of the metabolic aspects in immune regulation.
In the Macrophage section, I think the review will greatly benefit from making the statement in lines 132-133 and onward to analyze the data from single cell transcriptomics (for example, doi: 10.3389/fendo.2023.1198944) and other interesting analyses using this approach. The markers found in the M1 or M2-like lineages have been proposed. In this review, the authors can make their statement on what markers and cell states are the most accurate when characterizing the macrophages associated with metastatic states. In the specific sections for each type, the focus on metabolism in this reviewer's opinion, lacks other aspects such as iron acquisition and accumulation, which is very important for both the tumor and macrophage effector functions.
Another aspect that this reviewer thinks needs more depth is the role of iron in regulating the TME and the role in tumor killing.
Additionally, to the metabolic needs of immune cells in the TME, this reviewer thinks that authors could include some of the possible treatments to enhance immune system attack on tumors in hypoxic TME. As an example: doi: 10.3390/cancers15102738.
Finally, I think one topic that has not been discussed in depth in many reviews is the finding that cancer cells hijack mitochondria from immune cells; for example, doi.org/10.1186/1479-5876-11-94 and the study doi.org/10.1038/s41565-021-01000-4. The second study, I think, is particularly important since they demonstrated that using two inhibitors reduced the transfer of mitochondria. I strongly suggest incorporating this TME event as part of the metabolic complexity of cancer and the immune system.
In lines 632-634, I think this statement should be placed at the beginning of the manuscript, in my opinion, at the end of the introduction. Please consider this suggestion.

Additional comments

In lines 32-33, I think the authors should use "exhaustion". In this reviewer's opinion, the term reflects better the outcome of the reduced or lack of T cell response in the tumor microenvironment and the reduced effect of immune system stimulation.
I suggest using italics in "in vivo" in the manuscript.
For this reviewer, line 81 is unclear. Please rephrase.
Please, in line 306, add the alfa symbol to TNF-a.
In line 388, authors can use the abbreviations.
Please use superindex in CD8+ and CD4+ linages in the T cell section.
In line 391, please use a superindex in Calcium ions.
In line 397, please add the reference regarding the TME pH acidification.
In line 448, please remove the extra space in amino acid.
In this reviewer's opinion, in line 474, should read antibody therapy. Please revise.
In line 520, please reference the types of T cell-based immunotherapy.
Lines 526-527 should be revised; the years 2021 and 2022 have already passed, so the therapies are approved. Along the same lines, the toxicity found with CAR-T cells should be discussed and stated.
In lines 560-563, please discuss the pitfall of Metformin use, for example, the variation in dose in diabetic patients.

·

Basic reporting

The manuscript is clearly written in professional English. Literature references are more than sufficient and the background is provided. This review has a broad and interdisciplinary nature and meets the scope of this journal. It is very well written.

Experimental design

This is a review and as such it comprehensive, timely, current and in sightful.

Validity of the findings

This is an outstanding review of metabolic pathways myeloid cells and T cells in the tumor microenvironment (TME) utilize to promote or arrest tumor growth. Recent studies have clearly identified metabolic interactions between immune and non-immune cells in the TME that discriminate tumor from normal tissues. This review discusses metabolic pathways that drive M1 to M2 polarization and determine outcomes of the crosstalk between different subsets of T cells in the tME. In addition to a summary of various biochemical pathways involved in shaping the TME, the authors also provide suggestions for therapeutic interventions that could alter the immunosuppressive metabolic environment in the TME. The authors are careful to stress the bidirectional effects that characterize the metabolic pathways in the TME. Thus, caution and more research are needed before therapeutic alterations of the metabolic milieu in the TME can be considered.
Overall, this is a balanced and rational presentation of the TME which comprehensively summarizes recent progress in our understanding of multiple metabolic pathways that characterize it. The potential of therapeutic regulation of the metabolic pathways in the TME to benefit patients with cancer is clearly there, but it represents a considerable challenge and will require further investigation.

Additional comments

Excellent, detailed presentation of metabolic crosstalk in the TME.

Reviewer 3 ·

Basic reporting

The review of Zheng et al is an interesting review focused in macrophages and T cells in the tumor microenvironment and how the metabolic reprogramming of these cells may be the beginning of understanding the basis of immunotherapies. The review is clear with sufficient references and display appropriate structure.

I have included the comments below in chronological order to the manuscript for your consideration.

1. In paragraph: T cell dysfunction is caused by antigenic stimulation that lasts for a long time and is affected by things outside... I suggest to change things by elements or components.
2. I would suggest the authors to include percentages from cell types or populations cells in TME in introduction section.
3. Please include the words: in vitro, in vivo or ab initio in italics in the text.
4. In survey methodology section, please, include numerals to the subject terms.
5. In figure 1, please insert the word pH correctly.
6.In paragraph: In mouse models of cancer, macrophages promote tumor cell invasion, infiltration, and migration, facilitating cancer development and progression... I would suggest include or refer types of cancer in models of cancer.
7. In section of 1. M1-like TAMs, please include description of abbreviation PPP.
8. In figure 2, I would suggest include an arrow to represent increase or decrease metabolites in TCA or glycolysis for both M1 and M2-like macrophages.
9. According with paragraph: When adenosine binds to the A2A receptor, it stops macrophages from becoming M1-polarized during immune responses. On the other hand, when adenosine activates the A2B receptor, it makes macrophages become M2-polarized. I would suggest to be clear because A2B receptor is not in figure 2.
10. Please place homogeneously in the text the amino acid arginine because it appears in upper and lower case in the initial letter.

Experimental design

This review contain a correct survey methodology and it is mentioned that a total of 135 screened documents were finally reviewed. Please, include references 20-21 and 23-24 because sources are not quoted in the text. Also, please include pages number in all references.

Validity of the findings

No comment.

Additional comments

The review includes tables and figures to reinforce understanding of the text and summarise relevant information.

1. I would suggest that the metabolic pathway column of table 1 be sorted in alphabetical order.
2. Please, include the source of figure 1 if it is based on another reference.
3. In figure 2, please check that the edges of the arrows join together.
4. In figure 3, Ferroptosis is not explained in the text.
5. I would suggest adjusting the explanation for figure 3 because the text provides additional information that is not included in the figure and can be confusing.

·

Basic reporting

The review conducted by the authors is very detailed and provides a wealth of information regarding how the tumor microenvironment alters the metabolism of immune cells to facilitate tumor survival. Likewise, it describes how certain nutrients such as arginine, glutamine, and lactate can influence macrophage polarization, thereby modifying the inflammatory response and affecting tumor elimination or survival. Additionally, there is an excellent description of how the metabolism of T cells is also influenced by the tumor environment, promoting the generation of regulatory cells. This work contributes to understanding the metabolic state of T cells and macrophages in the tumor microenvironment, and the impact of metabolic reprogramming on tumor therapy will help optimize subsequent immunotherapy strategies.

Experimental design

This is a PubMed review of documents containing information on macrophage metabolism, macrophage, metabolic reprogramming, and immunotherapy, using various combinations of these terms. Duplicate works were excluded, and the information was compiled in the study, with a total of 133 documents analyzed.

Validity of the findings

No comment.

Additional comments

The review is thorough, and I would recommend incorporating a summary table to highlight the significance of key metabolites. This table could outline their influence on macrophage polarization and the functioning of other immune cells, including adenosine, lactate, glutamine, and arginine. Additionally, it would be valuable to explore the specific types of cancer where these metabolites exhibit a more pronounced impact, potentially serving as therapeutic targets for particular cancer types.

Figure 2, the left panel should clearly indicate the extracellular part, and there's potential for enhancement in this figure. Additionally, several definitions are missing, such as Teff and PPP. I suggest including these for clarity. It is common to use italics for terms like 'in vitro' and 'in vivo.' Figure 3 appears confusing; the interactions between different cells and the involvement of metabolites in tumor cell survival are not well-distinguished. It is unclear which metabolic pathways are inhibited or favored, and the information in the boxes is not easily comprehensible. It might be beneficial to separate the scenarios where Teff cells function against tumor cells, detailing the involved metabolic factors, and to present in an alternate figure the polarization towards regulatory T cells and the subsequent survival of tumor cells.

---

## Round 0.2 · accepted · Accept

The authors addressed all the raised concerns, and as a consequence, the manuscript is improved and ready for publication.